# Exploring physics of ferroelectric domain walls via Bayesian analysis of atomically resolved STEM data

Christopher T. Nelson [1], Rama K. Vasudevan[1], Xiaohang Zhang [2], Maxim Ziatdinov [1],
Eugene A. Eliseev [3], Ichiro Takeuchi[2], Anna N. Morozovska [4] & Sergei V. Kalinin [1✉]

The physics of ferroelectric domain walls is explored using the Bayesian inference analysis of atomically resolved STEM data. We demonstrate that domain wall profile shapes are ultimately sensitive to the nature of the order parameter in the material, including the functional form of Ginzburg-Landau-Devonshire expansion, and numerical value of the corresponding parameters. The preexisting materials knowledge naturally folds in the Bayesian framework in the form of prior distributions, with the different order parameters forming competing (or hierarchical) models. Here, we explore the physics of the ferroelectric domain walls in $BiFeO_3$ using this method, and derive the posterior estimates of relevant parameters. More generally, this inference approach both allows learning materials physics from experimental data with associated uncertainty quantification, and establishing guidelines for instrumental development answering questions on what resolution and information limits are necessary for reliable observation of specific physical mechanisms of interest.

[1] The Center for Nanophase Materials Sciences, Oak Ridge National Laboratory, Oak Ridge, TN 37831, USA. [2] Department of Materials Science and Engineering, University of Maryland, College Park, MD 20742, USA. [3] Institute for Problems of Materials Science, National Academy of Sciences of Ukraine, Krjijanovskogo 3, 03142 Kyiv, Ukraine. [4] Institute of Physics, National Academy of Sciences of Ukraine, 46, pr. Nauky, 03028 Kyiv, Ukraine. ✉email: sergei2@ornl.gov

Unique phenomena emerging at ferroelectric domain walls[1–7] have attracted the attention of researchers for many decades. Since the early days of ferroelectricity, it was recognized that the minimization of electrostatic energy of depolarization fields necessitates formation of ferroelectric domains, with the domain walls containing excess free energy compared to the bulk phase[8–12]. For several decades, the attention of the condensed matter physics and materials sciences communities alike was focused preponderantly on the domain properties and dynamics, whereas the walls were essentially treated as 2D objects. Correspondingly, even though remarkably advanced Ginzburg-Landau theory based theoretical models of wall structures were available as early as the late 1950s[13,14], these theories were experimentally unverifiable beyond macroscopic thermodynamic descriptors due to the lack of the high-resolution imaging tools capable of probing wall structures on the nanometer and atomic levels.

This situation has changed drastically in the last fifteen years, in which the improvements in characterization tools made such studies possible and the interest of physics community shifted to the intrinsic physics and applications of the ferroelectric domain walls. After the discovery of enhanced local conductivity at ferroelectric domain walls[1], tunable electronic properties have been demonstrated[4,15] and given rise to continuous research efforts towards domain wall electronics[2–4,15,16]. In conjunction with these experimental advances, a number of groups have theoretically explored the physics of the ferroic domain walls using the mesoscopic[17–19] and DFT models[20–23], and have demonstrated that suppression of the primary order parameter at the wall core can give rise to additional magnetic or polar functionalities[5,24–27]. The internal wall structure and hence conductivity are further strongly affected by the presence of flexoelectric interactions[28,29], and can thus be used to establish the strength of the latter[30]. In addition to purely physical functionalities, the domain walls were also shown to interact with the chemical subsystem in materials[31], giving rise to phenomena ranging from ferroelectric aging to vacancy segregation. While many of these theoretical advances suggest potential emergent physics at domain walls, experimental verification often remains a challenge even for atomic-scale, real-space imaging tools like (Scanning) Transmission Electron Microscopy (STEM).

Indeed, the equilibrium domain walls in proper ferroelectrics are usually very narrow, of the order of the atomic lattice parameter. Thus, STEM characterization of their internal structure has been enabled in the last decade by the introduction and proliferation of atomic-resolution spherical aberration corrected microscopes, allowing direct observations of the ferroelectric domain wall (and other interfaces) on the atomic level[32–42]. Quantitative information on the wall structure has been compared to Ginzburg–Landau–Devonshire (GLD) based models to yield materials parameters[40,43–45], and to validate DFT calculations[21]. However, despite the advance in imaging capabilities, the amount of material-specific information remains highly limited. Indeed, from early works of Ivanchik and Zhirnov it has been known that the order parameter profile across the domain wall is determined by the type of ferroelectric (second or first order), polarization behavior at the wall (Bloch, Ising, or Néel), and presence of the secondary order parameters or flexoelectric interactions[46,47]. Yet, while this information can theoretically be extracted from the wall profile, the experimental manifestation in the atomic structure can be subtle against instrumental noise or artifacts, is discretized at the level of atom positions, and is viewed in projection, precluding information from the 3rd spatial dimension (e.g., observation of Bloch character). This raises the statistical question of certainty in comparing multiple models to STEM profiles, or conversely, an estimation of the level of spatial resolution/information limit of the imaging system required to distinguish separate models.

Bayesian inference provides the probability of a model/hypothesis given a set of experimental observations, a powerful statistical technique rising in popularity with corresponding computational and statistical tools (we refer interested readers to more comprehensive texts such as refs. [48–50]). Here we develop this Bayesian inference framework for the analysis of material physics from structural imaging data. We demonstrate its application to the exploration of domain wall physics in the prototypical ferroelectric $BiFeO_3$ deriving the posterior probabilities for three GLD domain wall models from atomic resolution STEM experimental observations. Predicted domain wall shapes are dependent on parameters of the underpinning Landau theory. Under the assumptions of the validity of the theory, this then suggests that the domain wall profiles observed allow inversion to yield the parameters of the underlying Landau model, and thus, infer the order of the phase transition. We incorporate the effects of imaging noise and lattice discretization, and demonstrate how these can affect inferred materials properties. The required resolution limits to explore progressively fine details of domain wall physics are established, providing an answer to questions such as "how good of a microscope is necessary to address specific aspects of physics in a given materials system".

## Results
As the first step in this analysis, we discuss the relationship between the physics of the ferroelectric material and the order parameter profile across the domain wall. For multiferroic materials with the general form of the long-range order parameters, the wall profiles can be found using the classical LGD approach. Two vector long-range order parameters, polarization components $P_i$ and oxygen octahedral tilts $\Phi_i$, were used for the description of the antiferrodistortive (AFD), ferroelectric (FE), and antiferroelectric (AFE) long-range orders in the rare-earth doped $R_xBi_{1-x}FeO_3$, where R = Sm, La, Pr, Eu, etc.[27,51–53]. For completeness, we also add the antiferroelectric (AFE) long-range orders to the description. The bulk part of LGD thermodynamic potential consists of several contributions, which are listed in Supplementary Methods of Supplementary Information. For further explanations, we list only the compact form of the FE and AFE contributions as:

$$\Delta G_{FE} = a_i\left(P_i^2 + A_i^2\right) + a_{ij}\left(P_i^2 P_j^2 + A_i^2 A_j^2\right)$$
$$+ a_{ijk}\left(P_i^2 P_j^2 P_k^2 + A_i^2 A_j^2 A_k^2\right) + \gamma_{ij}^{ab}\left(P_i P_j - A_i A_j\right)$$
$$+ g_{ijkl}^{aa}\left(\frac{\partial P_i}{\partial x_k}\frac{\partial P_j}{\partial x_l} + \frac{\partial A_i}{\partial x_k}\frac{\partial A_j}{\partial x_l}\right) + g_{ijkl}^{ab}\left(\frac{\partial P_i}{\partial x_k}\frac{\partial P_j}{\partial x_l} - \frac{\partial A_i}{\partial x_k}\frac{\partial A_j}{\partial x_l}\right),$$

$$(1)$$

where the FE and AFE order parameters, $P_i = \frac{1}{2}\left(P_i^a + P_i^b\right)$ and $A_i = \frac{1}{2}\left(P_i^a - P_i^b\right)$, are introduced, $P_i^a$ and $P_i^b$ are the polarization components of two equivalent sublattices "a" and "b"[54–56]. As usual for proper and incipient ferroelectrics, the coefficients $a_k$ are temperature dependent and obey the linear law, $a_k(T) = \alpha_T[T - T_C]$, where $T_C$ is the Curie temperature, and $T$ is the absolute temperature; negative $a_k(T)$ supports FE or AFE state. The sign and value of $\gamma_{ij}^{ab}$ determines the AFE and FE phases coexistence.

For a general case of domain structured or spatially modulated system, one should solve the coupled Euler–Lagrange (EL) equations of states, which are expressed via the variational derivatives of the functional (1), $\frac{\delta G}{\delta P_i} = E_i$, $\frac{\delta G}{\delta A_i} = 0$. External and depolarization fields, $E_i^{ext}$ and $E_i^d$, which contribute to the electric field, $E_i = E_i^{ext} + E_i^d$, can be found from the electrostatic equation for electric displacement **D**, div **D** = 0, with boundary conditions

at the surfaces, interfaces and/or electrodes. Elastic fields, which are, in fact, the secondary order parameters, satisfy equation of state and mechanical equilibrium equations, whereas the strains and/or stresses should be defined at the system boundaries.

As a rule, the biquadratic coupling to the polar subsystem is small, and depolarization field is absent for uncharged walls. In this case, it is possible to reduce the non-local free energies to the decoupled system of equations for the order parameters:

$$2\left(a_i + a_{ij}P_j^2 + a_{ijk}P_j^2 P_k^2\right)P_i + \gamma_{ik}^{ab}P_k - g_{ijkl}^P \frac{\partial^2 P_j}{\partial x_k \partial x_l} \approx E_i, \quad (2a)$$

$$2\left(a_i + a_{ij}A_j^2 + a_{ijk}A_j^2 A_k^2\right)A_i - \gamma_{ik}^{ab}A_k - g_{ijkl}^A \frac{\partial^2 A_j}{\partial x_k \partial x_l} \approx 0, \quad (2b)$$

where $g_{ijkl}^P = g_{ijkl}^{aa} + g_{ijkl}^{ab}$ and $g_{ijkl}^A = g_{ijkl}^{aa} - g_{ijkl}^{ab}$ Since "aa" constants are for next-nearest neighbors sublattices, while "ab" constants are for the nearest neighbors sublattices, one can assume that, $\left|g_{ijkl}^{aa}\right| \ll \left|g_{ijkl}^{ab}\right|$, and so $g_{ijkl}^P \approx g_{ijkl}^{ab}$ and $g_{ijkl}^A \approx -g_{ijkl}^{ab}$. Here, we derive analytical solutions for Eq. (1) for several specific cases as described below.

**Model 1**. For the ferroelectrics with the second order phase transition the order parameter **P** across the uncharged domain wall can be found in the one component and one-dimensional approximations. Namely:

$$P_2(x_3) = P_S \cdot \tanh\left[\frac{x_3 - x_0}{L_c}\right], P_1(x_3) = 0, \quad (3a)$$

where $P_S^2 = -a_1/a_{11}$ is the spontaneous polarization, $x_3 - x_0$ is the distance from center of the domain wall plane, and $L_c = 2\sqrt{g_{44}/\left(a_1 + 3a_{11}P_S^2\right)}$ is the correlation length[57]. Eq. (3a) is valid at $a_1 < 0$, $a_{11} > 0$, $a_{111} = 0$ [see Fig. 1a, d].

**Model 2**. For the ferroelectrics with the first order phase transition, the order parameter profile is more complex:

$$P_2(x_3) = \frac{P_S \cdot \sinh[(x_3 - x_0)/L_c]}{\sqrt{\eta + \cosh^2[(x_3 - x_0)/L_c]}}, P_1(x_3) = 0, \quad (3b)$$

where $P_S^2 = \left(\sqrt{a_{11}^2 - 4a_1 a_{111}} - a_{11}\right)/2a_{111}$ and dimensionless parameter $\eta = 2a_{111}P_S^2/\left(3a_{11} + 4a_{111}P_S^2\right)$ is positive and its increase indicates the deviation from tanh-like profile (Eq. 3a). The correlation length is $L_c = 2\sqrt{g_{44}/\left(a_1 + 3a_{11}P_S^2 + 5a_{111}P_S^4\right)}$. Eq. (3b) is valid at $a_1 < 0$, $a_{111} > 0$ and arbitrary sign of $a_{11}$. Exact Eq. (3b) describe 180° Ising-type uncharged domain wall in uniaxial and multiaxial ferroelectrics [see Fig. 1b, e].

**Model 3**. For the ferroelectrics with the second order phase transition in the presence of possible polarization rotation, P-profile can be found for the specific case, $a_{12} = 6a_{11}$[46], and the solution is the superposition of two tanh-profiles:

$$P_2(x_3) = \frac{P_s}{2}\left[\tanh\left(\frac{x_3 - x_a}{\sqrt{2}L_c}\right) + \tanh\left(\frac{x_3 - x_b}{\sqrt{2}L_c}\right)\right]$$
$$\cong \frac{P_S \cdot \sinh[(x_3 - x_0)/L_c]}{\cosh[R_0/L_c] + \cosh[(x_3 - x_0)/L_c]}, \quad (4a)$$

$$P_1(x_3) = \frac{P_s}{2}\left[\tanh\left(\frac{x_3 - x_a}{\sqrt{2}L_c}\right) - \tanh\left(\frac{x_3 - x_b}{\sqrt{2}L_c}\right)\right]$$
$$\cong \frac{P_S \cdot \sinh[R_0/L_c]}{\cosh[R_0/L_c] + \cosh[(x_3 - x_0)/L_c]}, \quad (4b)$$

where $P_S^2 = -a_1/(2a_{11})$ is the spontaneous polarization ($a_1 < 0$, $x_3 - x_0$ is the distance from center of the domain wall plane $x_0 = \frac{x_a + x_b}{2}$,

the correlation length is $L_c = \sqrt{-g_{44}/(2a_1)}$, and $R_0 = x_b - x_a$ is an arbitrary constant. Exact expressions (4) describe rotational Ising–Bloch-type uncharged domain wall with Landau free energy density $g_{LGD} = \frac{a_1}{2}\left(P_1^2 + P_2^2\right) + \frac{a_{11}}{4}\left(P_1^4 + P_2^4\right) + \frac{a_{12}}{2}P_1^2 P_2^2 + \frac{g_{ijkl}}{2}\frac{\partial P_i}{\partial x_j}\frac{\partial P_k}{\partial x_l}$. [see Fig. 1c, f].

The ranges of the possible values for the 4 parameters in Eqs. (3–4) are $P_S = (0.1–1)$ C/m², $L_c = (0.5–5)$ nm, $\eta = (0–100)$, and $R_0/L_c = (0–10)$. Experimentally, an additional variable is the wall position, $x_0$. Note that within the LGD approach, the "true" independent parameters are the coefficients in the GLD expansion, $a_1$, $a_{11}$, $a_{111}$, and $g_{44}$. However, for the analysis of the experimental data we treat the phenomenological wall parameters as independent.

Note that similar analysis can be performed for more complex physical mechanisms, albeit in these cases the numerical solutions are generally required. Further note that Model 1, Eq. (3a) is a special case of Model 2, Eq. (3b) for $\eta = 0$, as well as of Model 3, Eqs. (4) at $R_0 = 0$, as it can be expected from the physics of the problem. At the same time, Models 2 and 3 are alternative models, corresponding to the dissimilar physics of the material.

While the solutions (Eqs. 3–4) are limited for specific numerical values of free energy expansion, finite element analysis (FEM) confirms that a direct variational method with slightly more complex trial functions can be used to describe the rotation domain wall at $a_{12} \neq 6a_{11}$:

$$P_2(x_3) = P_a \tanh\left(\frac{x_3 - x_a}{a}\right) + P_b \tanh\left(\frac{x_3 - x_b}{b}\right), \quad (5a)$$

$$P_1(x_3) = P_a \tanh\left(\frac{x_3 - x_a}{a}\right) - P_b \tanh\left(\frac{x_3 - x_b}{b}\right)$$
$$+ P_c\left[1 - \mu \cosh^{-2}\left(\frac{x_3 - x_c}{c}\right)\right], \quad (5b)$$

where $P_{a,b,c}$, $x_{a,b,c}$, $a$, $b$, $c$, and $\mu$ are variational parameters.

These analyses set the context for the problem of physics determination from experimental data. Namely, the question we seek to answer is in which cases the more complex behaviors defined by Model 2, Eq. (3b), can be reliably differentiated from the simplest behavior of Model 1, Eq. (3a), thus differentiating materials with first and second order phase transitions. Similarly, how well can Models 1, 2, and 3, or the more complex models in Eq. (5), be separated given the noise level of the measurement system and the discretization in measurements induced by the underlying lattice. How can prior knowledge of the material system be used to narrow down the answers? And finally, given the possible range of materials properties, can we establish the requirements on microscope resolution/information limit required for these to be determinable?

The answers for these questions can be naturally explored in the context of Bayesian inference. Bayes formula relates the prior and posterior probabilities as

$$p(\theta_i|D) = \frac{p(D|\theta_i)p(\theta_i)}{p(D)}, \quad (6)$$

where $D$ represents the experimental data, $p(D|\theta_i)$ is the likelihood of the data given the model, $i$, with model parameters, $\theta_i$, and $p(\theta_i)$ is the prior, i.e., probability of the model. Finally, $p(D)$ is the denominator that defines the total possible outcomes.

Despite the long history of Bayesian theory, its adoption by the basic science fields has been slow. Part of the reason is that only a few special distribution classes allow analytical solutions of Eq. (6), whereas many realistic model classes require numerical methods and extremely cumbersome integrations. Secondly, the classical argument against a Bayesian approach is the need for the prior distributions, and dependence of the answers on the priors. Here we note that while problemic in medicine, sociology, or

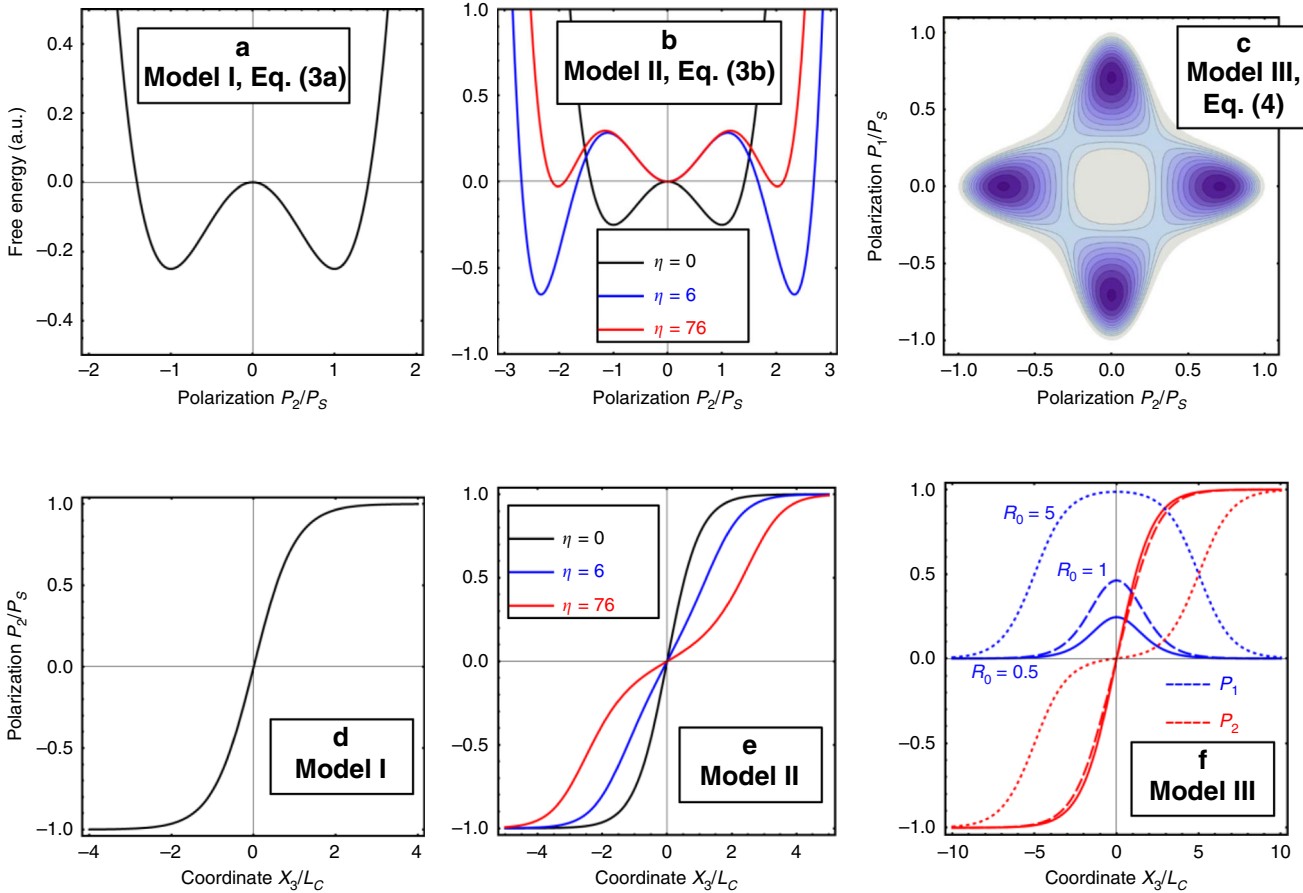

**Fig. 1 LGD ferroelectric domain wall models. a–c** Free energy vs. order parameter–polarization components $P_i/P_S$ and (**d–f**) distribution of these $P_i/P_S$ across the domain wall for Models 1–3. Note that we fixed the coefficients $a_1$, $a_{11}$, and $a_{111}$, and recalculated the spontaneous polarization $P_S$ and parameter $\eta$ from them for Model 2. For Model 3, component $P_1$ is not observable by STEM.

economics, in physical areas domain knowledge is often sufficiently developed to provide meaningful priors, as explored below. The main point to note here is that in Bayesian inference we are aiming to compute the posterior distributions of the parameters of some model, conditioned on data, which is usually done with an appropriate sampling method such as the Metropolis algorithm. This allows one to numerically estimate the posterior in (Eq. 6) for all model parameters. Note that the variance in the data can itself be modeled by a parameter, as noted below.

Here, we formulate a Bayesian regression models based on Models 1, 2, and 3. We note that STEM data does not provide an absolute calibration for polarization magnitude and the wall position is a priori unknown. Correspondingly, we chose weakly informative priors for these parameters. Similarly, for a second order ferroelectric described by Eq. (3a), the correlation length, $L_c$, is the sole parameter defining wall structure, and hence the corresponding prior can also be weak. For a first order ferroelectric described by Eq. (3b), the correlation length, $L_c$, and $\eta$ are parameters to be inferred. Note that model (3a) is the special case of (3b) for $\eta = 0$, and hence the target of Bayesian analysis is to establish whether $\eta$ is practically equivalent to zero (and hence the material is second order), or nonzero, and hence the material is first order. Finally, model Eq. (4a) has a different functional form than Eq. (3b), and hence determination of the parameters $L_c$, $R_0$ and separation of models 2 and 3 is the task for Bayesian inference.

The behavior of the posterior distributions was extensively explored using synthetic data made via a-priori known models

**Table 1 Priors used in sampling.**

| Model 1, Eq. (3a) | | Model 2, Eq. (3b) | | Model 3, Eq. (4a) | |
|---|---|---|---|---|---|
| $x_0$ | ~Uniform(0,L) | $x_0$ | ~Uniform(0,L) | $x_0$ | ~Uniform(0,L) |
| $P_S$ | ~Uniform(0.5 $P_{est}$, 2 $P_{est}$) | $P_S$ | ~Uniform(0.5 $P_{est}$, 2 $P_{est}$) | $P_S$ | ~Uniform(0.5 $P_{est}$, 2 $P_{est}$) |
| $L_C$ | ~Uniform(0,5) | $L_C$ | ~Uniform(0,5) | $L_C$ | ~Uniform(0,5) |
| var | ~HalfNormal(1) | var | ~HalfNormal(1) | var | ~HalfNormal(1) |
| $P_0$ | ~Uniform(−1,1) | $P_0$ | ~Uniform(−1,1) | $P_0$ | ~Uniform(−1,1) |
| | | $\eta$ | ~HalfNormal (10) | $R_0$ | ~HalfNormal(10) |

Ranges are shown in parenthesis, $L$ is size of the image in unit cells, $P_{est}$ is estimated saturated polarization.

(see Python notebook Supplementary Data 1) as a function of parameter values, noise level, and digitization step. It was established that for very thin domain walls with the thickness comparable with the lattice spacing, the lattice discretization becomes the most limiting factor in the analysis. Correspondingly, the inference allows only to distinguish the main features of the observed physical behaviors. Hence here as priors, we choose weakly informative priors for all associated parameters, as shown in Table 1.

Domain walls in the rhombohedral proper-ferroelectric archetype BiFeO$_3$ were taken as the model system for this analysis. A pseudocubic (pc) perovskite, BiFeO$_3$ adopts spontaneous polarization spatially degenerate along the eight <111>$_{pc}$

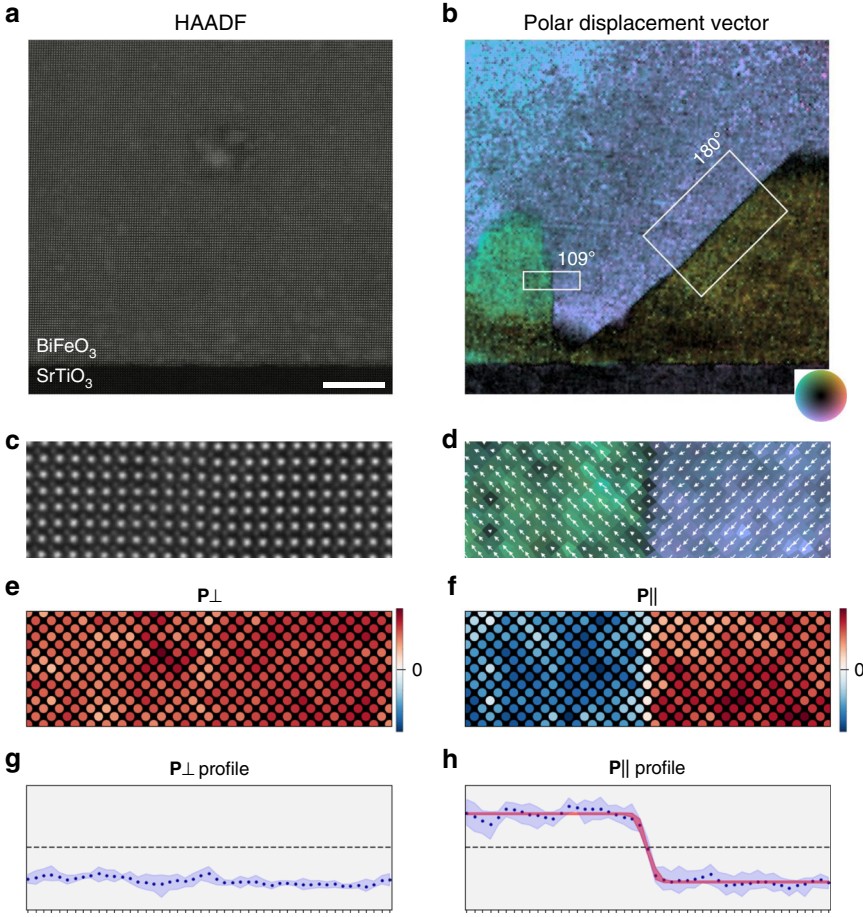

**Fig. 2 Experimental data for full-dataset and 109° domain wall. a, b** HAADF-STEM of polydomain BiFeO$_3$ thin film with magnified (**c–h**) 109° domain wall region. **a** HAADF-STEM of BiFeO$_3$ film on SrTiO$_3$ substrate, scalebar 10 nm. **b** Cation polar displacement vector, **P**, map, white rectangles depict subregion datasets bisecting 109° and 180° domain walls. **c** Subregion 109° domain wall in HAADF and (**d**) displacement vector maps. **e** Color-scaled lattice positions of the **P** component perpendicular ([100]$_{psuedocubic}$ axis) and (**f**) parallel ([001]$_{pc}$) to the domain wall. **g** Profiles of the mean values (datapoints) and 90% data bounds (blue) for perpendicular and (**h**) parallel **P** components. The red band corresponds to the 90% highest posterior density interval for the Bayesian analysis Model 2.

directions with three possible rotation angles between adjacent domains: 71°, 109°, and 180°. A BiFeO$_3$ thin film was grown on a SrTiO$_3$ substrate by pulsed laser deposition, the electrostatic energy from the insulating substrate interface promoting polydomain formation. A cross section of the film was imaged along the <100>$_{pc}$ direction by high-angle annular dark field (HAADF) STEM, producing a mass-thickness sensitive picture of the atomic columns (Fig. 2a). This does not provide a direct measure of electric polarization, however BiFeO$_3$ exhibits a strong lattice coupling in the form of non-centrosymmetric Bi-site displacements, which can be used as a proxy, and we hereafter refer to them with the same **P** notation. This cation polar displacement was calculated for the 4-atom nearest neighbors after a 2-orthogonal image scan-artifact reconstruction[58] and Gaussian fitting. A colorized vector map of this polar displacement is shown in Fig. 2b, clearly illustrating the polydomain structure of the film. In this case, typical equilibrium 109° and 180° domain walls are found forming on the [100]$_{pc}$ and [$\bar{1}$01]$_{pc}$ planes, respectively. Subregions indicated by the white boxes are used as the input experimental data for Bayesian network analysis.

A magnified view of the 109° domain wall from the region of interest (ROI) in Fig. 2b is shown in Fig. 2c, d, the boundary dictated by kinks in the domain wall. The change in the [010]$_{pc}$ component is not visible as it is along the viewing direction,

leaving a project 90° rotation corresponding to a transition in the [001]$_{pc}$ component as seen in the overlaid vector field in Fig. 2d. A lattice coordinate space is utilized for subsequent analysis, defining the normal distance from the domain wall in lattice parameter units. The polarization components are defined in reference to the domain wall plane, the perpendicular (P[100]$_{pc}$) and parallel (P[001]$_{pc}$) axes for the 109° domain wall. A lattice-coordinate spatial plot of these two components (Fig. 2e, f) and view of their profile data (Fig. 2g, h) illustrate the polarization transition occurring parallel to the plane. This component is used as the input for the Bayesian network analysis.

The model parameters for the 109° domain wall are shown in Fig. 3. The 90% highest posterior density interval is illustrated in the vicinity of the domain wall for all three models, overlaying mean values of the experimental data for each unit lattice distance, Fig. 3a. A least squares fit curve is also shown along with corresponding fit parameters. However, the power of the Bayesian analysis over a minimum value optimization is the probability information it provides. The posterior probabilities for the wall position and saturation polarization show near Gaussian distributions, allowing localized wall position and the bulk polarization, and an estimate of their associated errors. The use of strongly informative priors, e.g., constraining polarization to almost constant values, leads to the nearly uniform posterior

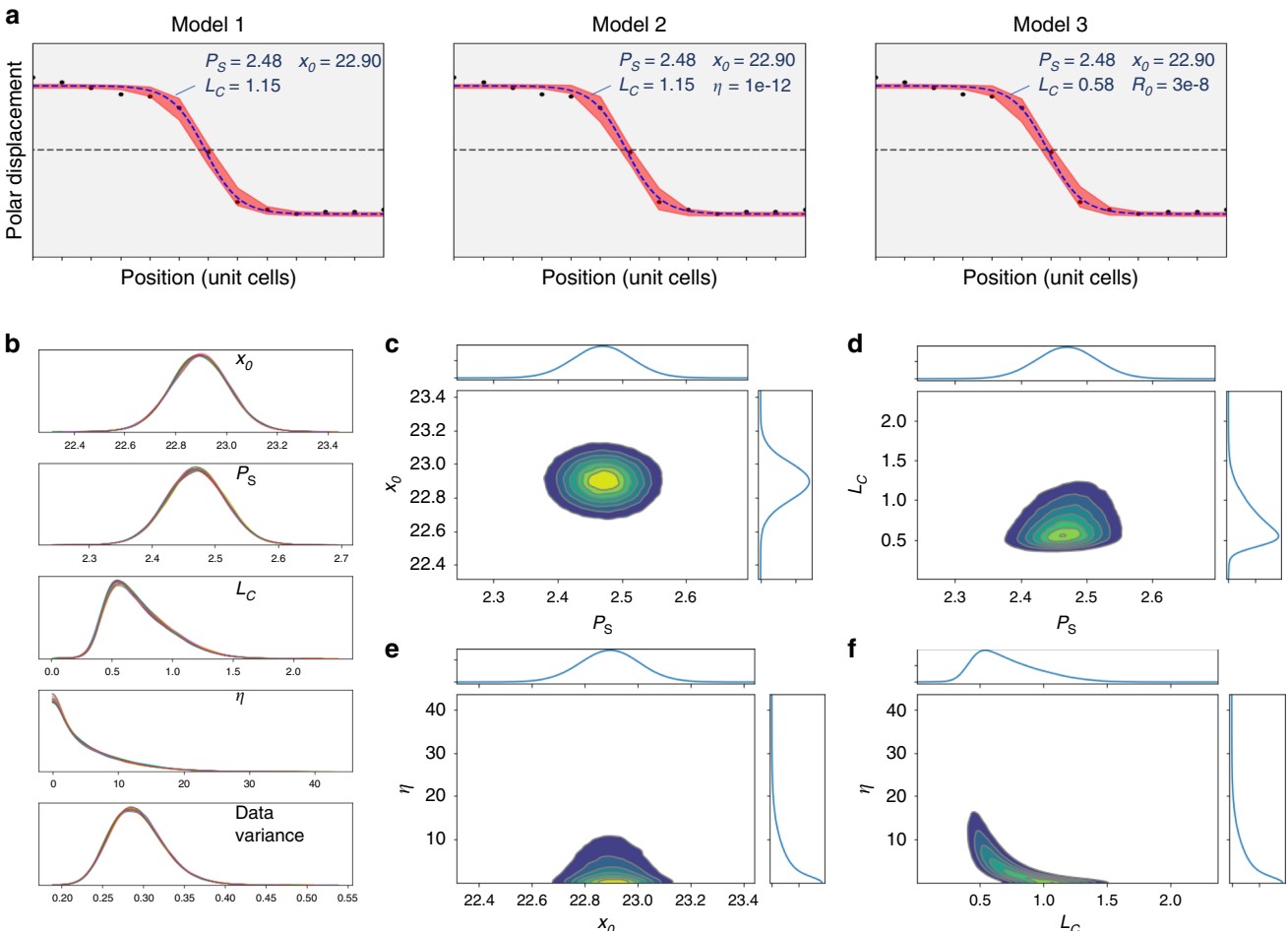

**Fig. 3 109° Domain wall: LGD models and posterior probability densities. a** The 90% highest posterior density interval for the Bayesian analysis (red band) overlaid on experimental mean values (data points). Dashed blue line is a least square fit. **b** Posterior probability densities (PPD) for the Model 2 parameters, Eq. (3b). Shown in **b** are 1D PPD for wall position, saturation polarization, correlation length, $\eta$, and the data variance. **c**–**f** are selected 2D joint probability densities for different parameter combinations. Some parameters (e.g., **c**, **d**, **e**) are clearly marginalizable, whereas the parameters $L_c$-$\eta$ (**f**) show that probability density is not marginalizable. These joint distributions illustrate how knowledge of one parameter can significantly enhance the estimate on the other.

densities and hence was avoided here. The posterior distributions for $L_c$ and $\eta$ show considerably more interesting behavior. The distribution for the $L_c$ is skewed towards large values, with the 3–97% Probability Density Function (PDF) being in the range (0.35–1.21). The corresponding $\eta$ distribution has the PDF range (0.0–15.0), with the clear maximum at $L_c$~0.5.

Further insight into the wall behavior can be inferred from the selected pairwise joint probability distributions as shown in Fig. 3b–d. Here, for most parameters the pairwise distributions are clearly marginalizable, i.e., the joint probability distribution can be well approximated as a product of the marginal distributions for individual parameters. This behavior implies that the variables are statistically independent, and the knowledge of one variable does not improve understanding of the other one. This behavior is clearly shown for wall position and polarization, $P_s$-$x_0$, pair. Similarly, $P_s$-$L_c$ and $x_0$-$\eta$ are close to marginalizable.

At the same time, the posterior distribution for the $L_c$ and $\eta$ shows very different behavior. The joint distribution is not marginalizable and shows strong parameter dependence. This implies that the knowledge of one of the parameters can significantly affect the amount of information we learn about the other one. As an example, the fit with the narrow parameter for $L_c$ in the (0.9–1.1)

interval is compared to the elements in Fig. 3b–f in the Supplementary Fig. 1. Other combinations of the model parameters can be explored using the notebook provided as Supplementary Data 1.

This analysis clearly illustrates the natural way in which the Bayesian inference allows to explore the available data given the past knowledge of the physics of the system. We have further performed the analysis for the third model Eq. (4a) and associated parameters and distribution functions are provided in the Supplementary Data 1.

To compare the models, we apply the widely applicable information criteria (WAIC)[59]. Most model selection criteria are based on two terms: one term that describes how well the model fits to the observed data, and a second term that penalizes models with greater degrees of freedom. As such, the WAIC is equal to (LL–$p_{WAIC}$), where

$$\text{LL} = \sum_{i=1}^{n} \log\left(\frac{1}{S}\sum_{s=1}^{S} p\left(y_i | \theta^s\right)\right)$$

$$p_{WAIC} = \sum_{i=1}^{n} \text{var}_{post}\left(\log p\left(y_i | \theta\right)\right)$$

where LL is the log likelihood, which is calculated by drawing $S$ samples from the posteriors of the parameters of a model, and the second term represents the effective number of parameters of the model, which is calculated by summing the variance of the log-likelihood. This is done for each model to compute the WAIC, and then the weights are calculated using a pseudo-Bayesian model averaging approach with AIC-type weighting[60]. The WAIC scores for the models are shown in Table 2. Based on the Bayesian analysis Model 1, the second-order ferroelectric, is the most likely. However, the relative probabilities of all three models are very close and demonstrate that given the observations for weakly informative priors the models cannot be distinguished. We argue (and confirm later) that this indistinguishability of the models in this case is due to the small domain width (compared to lattice spacing), which precludes the subtle differences in domain wall structure between models from being reliably identified.

We repeat this analysis for the case of the 180° domain wall ROI in Fig. 2a, the experimental data shown in Fig. 4. The 180° domain wall corresponds to a reversal of the polarization along the same axis, also appearing as a 180° transition in $[010]_{pc}$ projection, albeit the $P[010]_{pc}$ component is again not being observed. The polar displacement is again broken into perpendicular and parallel components. For the $[\bar{1}01]_{pc}$ domain wall, this corresponds to $[\bar{1}01]_{pc}$ and $[101]_{pc}$, respectively, and the latter is used as the input to the Bayesian analysis. There is an observable asymmetry with **P** not centered around zero, a known artifact of small off-axis mistilts which manifests in a similar polar cation asymmetry[61]. Since we are concerned with the polarization delta of the domain wall, this component is treated with a fixed offset term, $P_0$, added to the fitting function.

The model parameters for the 180° domain wall are shown in Fig. 5. The 90% highest posterior density interval for the three models and least squares fits (Fig. 1a) are, like the 109° case, very similar, with Models 2 and 3 adopting parameters that approach Model 1, the second-order ferroelectric Eq. 3a. WAIC scores for the three models again also suggest Model 1 is most likely (Table 2). Selected pairwise joint probability distributions from Model 2 are shown in Fig. 5b-g with results similar to the 109° case, albeit with an additional $P_0$ parameter. $P_0$–$P_S$ (Fig. 5b), $x_0$–$P_S$ (Fig. 5c), and $\eta$–$x_0$ (Fig. 5f) are clearly marginalizable, whereas $L_c$–$P_S$ (Fig. 5d), and $P_0$–$x_0$ (Fig. 5e) are somewhat close, and $\eta$–$L_c$ (Fig. 5g) is not. Although parameter correlations may not be linear, and clearly aren't in cases such as $\eta$–$L_c$, Pearson correlation coefficients are helpful to highlight this distinction, with $P_0$–$P_S$ (0.11), $x_0$–$P_S$ (−0.04), $\eta$–$x_0$ (0.14), $L_c$–$P_S$ (0.20), $P_0$–$x_0$ (−0.27), and $\eta$–$L_c$ (−0.80).

The analysis above illustrates the determination of the physically relevant parameters of the material given the experimental observations as STEM atomic coordinates, and past knowledge in the form of the Bayesian priors on relevant materials parameters. As expected for ferroelectric materials with the extremely narrow domain walls, ultimately this consideration becomes the limiting factor in these studies. In other words, while domain wall shape is a measure of the physics of the order parameter in the system, practically the STEM observations are limited by the discreteness of the lattice and the noise in the system.

To explore the effect of the noise and sampling on the differentiability of specific physical behaviors from the observational data, we perform a range of numerical experiments on synthetic data. Here, the synthetic data set is generated using Model 2, Eq. (3b), for a first order ferroelectric with a specific set of ground truth parameters, chosen here to be $(P_s, L_c, \eta) = (0.5, 2.0, 2.0)$. Varied parameters are noise level, chosen to be Gaussian with the dispersion $\sigma$, and number of sampling points in the interval, $N$, with the $x$ varying from −15 to 15. Correspondingly, the pixel spacing $30/N$ provides the measure of the discreteness of the measurements, and should be compared to the domain wall width, controlled by $L_c$ and $\eta$. The center position of the wall is chosen always at $x_0 = 0$ and there is no offset $P_0$. The generated synthetic data set is fit by the Bayesian model corresponding to Model 1, Eq. (3a), and Model 2, Eq. (3b), for a range of $N$ and $\sigma$ values. The point estimates of the recovered domain wall parameters $\hat{P}_s$, $\hat{L}_c$, $\hat{\eta}$, and $\hat{\sigma}$ are determined and can be compared with the ground truth values. Similarly, the WAIC score for the Models 1 and 2 can be determined.

**Table 2 WAIC model comparison for the 109° and 180° domain walls.**

| Model | 109° wall | | 180° wall | |
|---|---|---|---|---|
| | Rank | Weight | Rank | Weight |
| Model 1 | 1 | 0.49 | 1 | 0.56 |
| Model 2 | 3 | 0.22 | 2 | 0.11 |
| Model 3 | 2 | 0.29 | 3 | 0.33 |

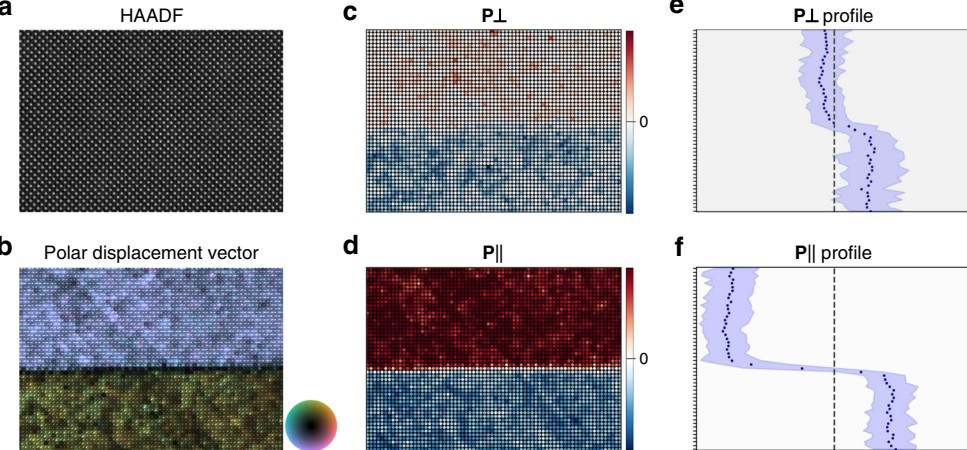

**Fig. 4 Experimental data for 180° domain wall. a** HAADF-STEM subregion of 180° BiFeO$_3$ domain wall. **b** Cation displacement vector map. **c** Color-scaled lattice positions of the **P** component perpendicular ($[\bar{1}01]_{pc}$ axis) and (**d**) parallel ($[101]_{pc}$ axis) to the domain wall. **e** Profiles of the mean values (datapoints) and 90% data bounds (blue) for perpendicular and (**f**) parallel **P** components.

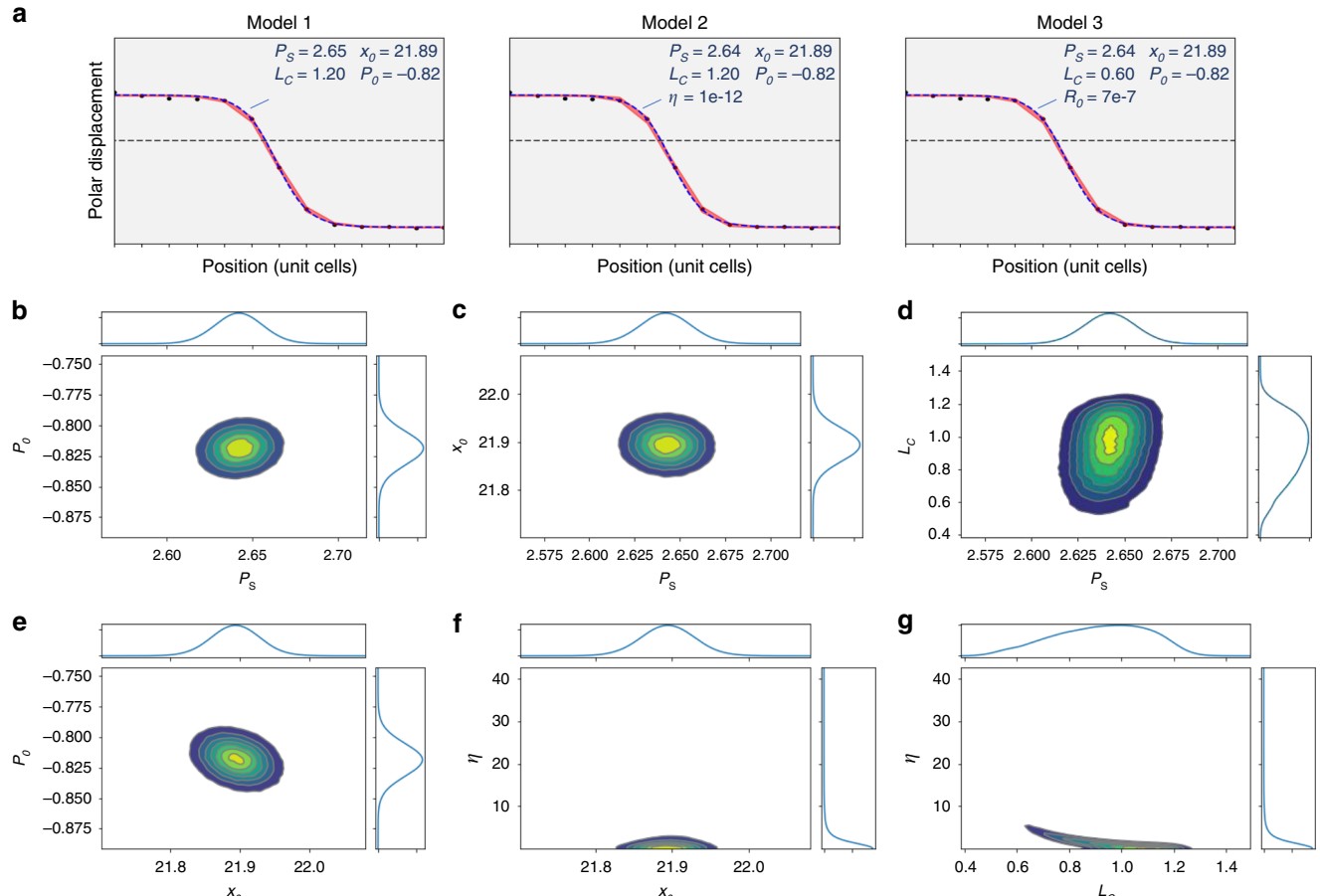

**Fig. 5 180° Domain wall: GLD models and posterior probability densities. a** The 90% highest posterior density interval for the Bayesian analysis (red band) overlaid on experimental mean values (data points). Dashed blue line is a least square fit. **b–g** are selected 2D joint probability densities for different parameter combinations for model 2.

Shown in Fig. 6 is the simulated domain wall profile for $N = 40$ and noise level $\sigma = 0.001$, 0.03, and 0.1. Weight values from comparison of WAIC scores from Models 1 and 2 as a function of noise level are shown in Fig. 6b. Here, 0 corresponds to identification of the correct Model (#2) of the generated data. The correct physical model can be determined from data only for noise levels below ~0.02. For higher noise values the weight centers round 50% (chance), and the model cannot be established from experiment alone. However, as shown in Supplementary Fig. 1, if the model and its parameters are partially known, they can be used as physics-based prior to determine materials properties more precisely.

The corresponding inferred parameters are shown in Fig. 6c–e. Here, in all cases the inferred noise level $\hat{\sigma}$ is very close to the ground truth level, $\sigma$. The point estimates of the domain wall parameters $\widehat{P}_s$, $\hat{L}_c$, $\hat{\eta}$ coincide with the ground truth values for small noises, and start to deviate for large noise level (above ~0.01–0.02). For parameters $\widehat{P}_s$ and $\hat{x}_0$ (reconstructed wall position), the uncertainty grows ~linearly with the noise, as can be expected from the functional form of the model. For parameters $\hat{L}_c$ and $\hat{\eta}$ the dependence is more complicated, and particularly for $\hat{L}_c$ the inferred value is centered at the ground truth value and is weakly affected by noise.

Finally, we explore the combined effect of the sampling and the noise level on the separability of the physical models given experimental data and only weak physics-based priors. In this approach, we aim to answer the fundamental questions as to

what level of microscope resolution/information limit is required to reliably determine the generative physics model from the data. Note that the issue of the information limit in the measured atomic positions and its relationship to microscope parameters is not explored here and we refer to other publications where these studies are performed[62–64]. Here, we seek to explore only the question as to which extent the correct (here, a priori known) physical model can be determined from experimental data given the sampling induced by lattice, and to which extent uncertainty in atomic positions (here, noise) affect this inference. However, we do not address the origin and mitigation strategies for the uncertainties for atomic positions,

To explore this issue, we introduce the ground truth model corresponding to Model 2 with $(P_s, L_c, \eta) = (0.5, 2.0, 2.0)$. We further create the calculated profiles at different samplings and noise levels. Bayesian inference is used to calculate the WAIC for Models 1 and 2. Where the WAIC comparison weight is close to 0, the models can be reliably separated, i.e., the physics of material can be established from the data. Where the value is 0.5 or larger the correct model cannot be determined from the data. The modeling results are shown in Fig. 7, where for convenience the units for pixel spacing are chosen to be comparable to domain wall width, i.e., $30/(N L_c)$. Figure 7a clearly illustrates that models can be distinguished as long as pixel size is comparable to the wall width, with the threshold value ~1.9. At the same time, for the noise levels above 0.02 the

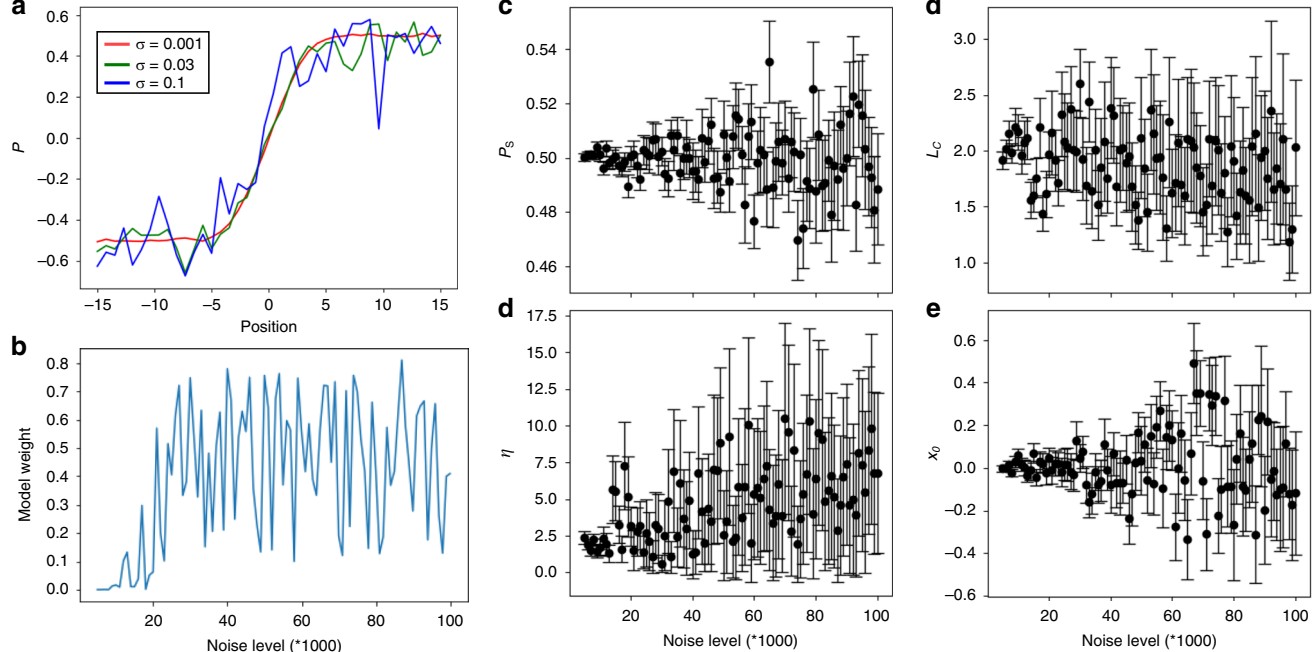

**Fig. 6 Noise and sampling effects. a** The simulated domain wall profiles for different noise levels. **b** Weight of Model 1 and 2 comparison of WAIC scores versus input noise. Zero value corresponds to correct identification of the wall type. **c–e** The extracted parameters from the Bayesian inference fits for single realization of the synthetic data set versus input noise.

model cannot be established. This threshold seems to be only weakly dependent on the pixel size.

We further explore the effect of prior physical knowledge on physics extraction. Figure 7b, c illustrates the effect of transition from weak to strong physical priors, where distributions of possible parameter values are much better defined. During the inference, such strong priors tend to produce uniform posteriors or posteriors sharply concentrated on the boundary or interval, as opposed to the Gaussian-like posteriors for weak priors. Note that the effect of strong prior can be roughly compared to the three-fold reduction in the noise level (compare Fig. 7a, c). Notably, the absolute knowledge of priors (Fig. 7c) does not considerably improve analysis compared to strong priors (Fig. 7b). However, incorrect priors, Fig. 7d, has strongly deleterious effect on analysis, effectively precluding model inference for all but extremely high-quality data. Overall, the additional physical knowledge can provide significant improvement in the analysis. However, incorrect knowledge provides a much stronger effect, calling for care with analysis.

## Discussion

To summarize, the physics of ferroelectric domain walls in BFO is explored via Bayesian inference analysis of atomically resolved STEM data. This approach allows for determination of materials parameters in the form of a relevant posterior distribution, based on prior materials knowledge in the form of prior distributions and available experimental data. The Bayesian inference can further be extended to analyze the likelihood of alternative models for materials physics. Here, we show that for non-informative or weakly informative priors (equivalent to classical point estimates in least-square fits) we can establish the model parameters and their posterior distributions, as well as attempt to distinguish the models. For the specific case of 109° and 180° domain walls in BiFeO$_3$, the combination of sampling and noise preclude a reliable differentiation of physical mechanisms.

However, incorporation of materials knowledge in the form of prior distributions of parameters can significantly narrow posterior distributions and allow for differentiation of generative mechanisms.

As a future perspective, we note that rather than using analytical models, Bayesian inference can be based on numerical solvers or even atomistic methods. However, this approach will considerably increase computational complexity, and may require approximate models based on Gaussian Processing[48,50] or deep learning-based interpolators. Similarly, this approach can be applied to other physical models, including those with more complex order parameters, in the presence of electronic or ionic screening. This includes non-equilibrium cases, e.g., strain during preparation or heating profile, as long as the numerical schemes for forward modeling are available.

More generally, Bayesian methods allow for a natural framework to distinguish possible physical mechanisms in observations, and allows us to very clearly ascertain to what extent we learn more from knowledge of the microscope and materials. This analysis clearly illustrates that additional physics or knowledge leads to the increase of the value of physical experiment. However, incorrect knowledge can strongly obviate any potential information gain. Taken over the multiple domains, it provides clear and quantifiable stimulus towards development of high-resolution microscopies and high information content probes such as four-dimensional (4D) STEM, and allows exploring cost-benefit considerations in the microscope development.

## Methods

FE domain wall models are based on the Ginzburg–Landau–Devonshire (LGD) approach, detailed in Supplementary Methods of Supplementary Information

The Bayesian inference is implemented via the PyMC3 library in Python[59]. Weakly informative priors were chosen consisting of uniform distributions for wall position, polarization, and correlation length and half-normal distributions for $\eta$ and data variance. The Metropolis Monte-Carlo algorithm was used with 5 k tuning steps, 50 k computational steps, and 16 chains. The total computation time

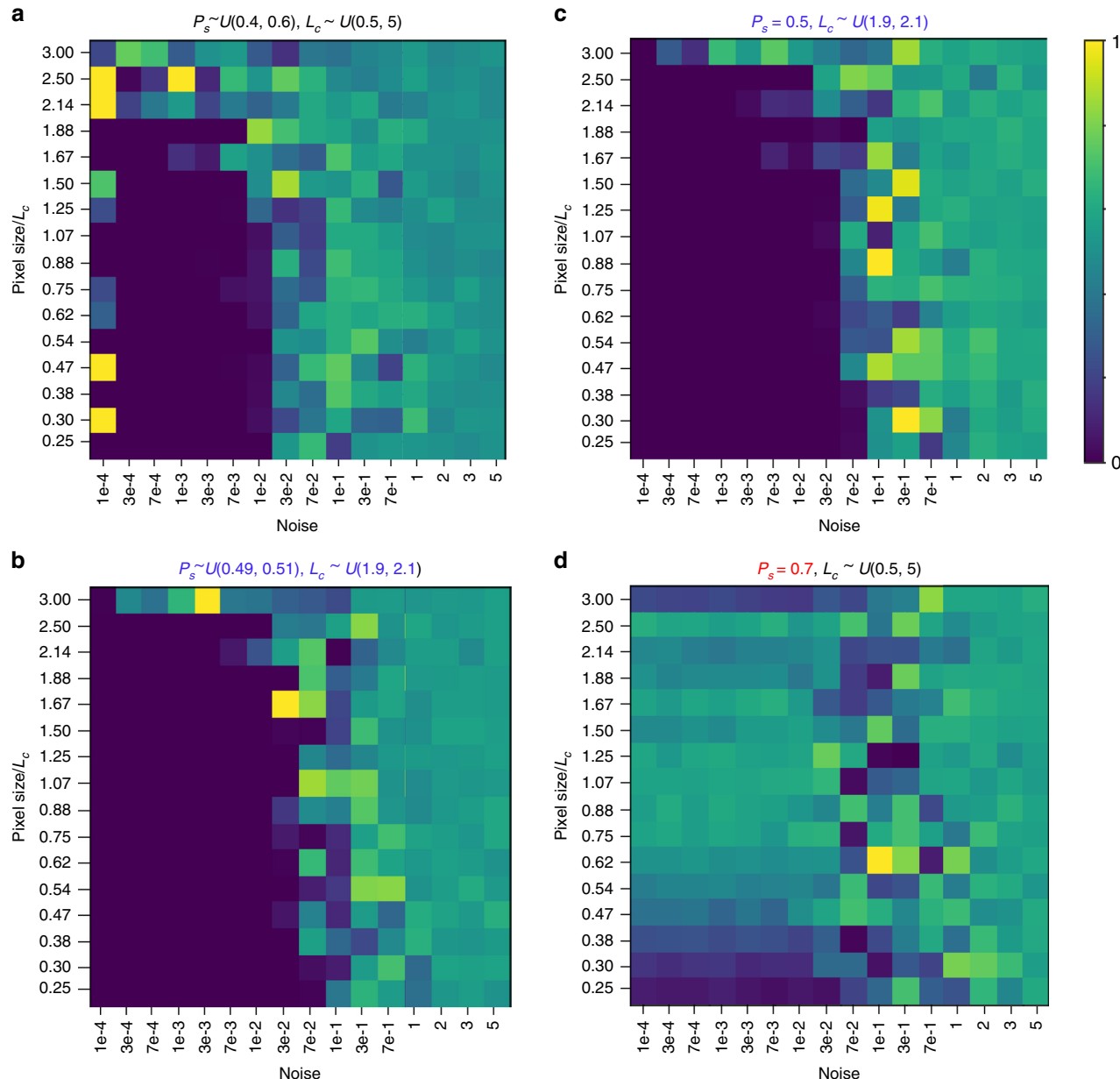

**Fig. 7 Effect of prior knowledge on imaging required for model separation.** Matrices show the weight for WAIC score comparisons of Models 1 and 2 against synthetic domain wall data as a function of the effective pixel size and noise level. Universal color scalebar is at right of the image, 0 corresponds to models being reliably separated, 0.5 is indistinguishable, and 1 is incorrect inference. The true value of parameters are $x_0 = 0$, $P_s = 0.5$, $L_c = 2$, and $\eta = 2$. The prior distribution of parameters are labeled, black text indicates comparison values, blue represents additional knowledge that allows more informed prior, and red represents incorrect knowledge. **a** Weakly informed priors on polarization and correlation length. **b** Strongly informed priors. **c** Exact polarization and weakly informed correlation length priors. **d** Incorrect polarization and weakly informed correlation length priors.

on Google Colab is ~12 min. The joint posterior probability densities are visualized using Arviz library. Select posterior densities are shown in Figs. 3, 5, and Supplementary Fig. 1. The analysis presented in this work, posterior densities, Markov–Chain Monte–Carlo traces, and plotting methods are available in the accompanied Python notebook Supplementary Data 1.

Epitaxial $BiFeO_3$ thin films were synthesized on (001) $SrTiO_3$ substrates by Pulsed Laser Deposition at a laser fluence of ~0.8 J/cm$^2$, growth temperature of 600 °C, and deposition pressure of ~100 mTorr from Oxygen flow after a base pressure of ~$2 \times 10^{-8}$ Torr.

The STEM sample was fabricated by a Focused Ion Beam cross-sectional liftout and subsequent Argon ion milling in a Fischione NanoMill with a final energy of 0.5 keV. STEM was performed on a NION UltraSTEM operating at 200 keV, the data utilized here corresponding to the High-Angle Annular Dark Field (HAADF) detector. Cation polar displacements were calculated for the 4-atom nearest

neighbors after a 2-orthogonal image scan-artifact reconstruction[58] and Gaussian atom-fitting.

## Data availability

The supporting data is publicly available in a Zenodo repository with identifier https:// doi.org/10.5281/zenodo.4122471.

## Code availability

Analysis code for this work is available in a Python notebook as Supplementary Data 1.

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

## Acknowledgements
This effort (electron microscopy, Bayesian inference) is based upon work supported by the U.S. Department of Energy (DOE), Office of Science, Basic Energy Sciences (BES), Materials Sciences and Engineering Division (S.V.K., C.T.N., R.K.V.) and was performed and partially supported (M.Z.) at the Oak Ridge National Laboratory's Center for Nanophase Materials Sciences (CNMS), a U.S. Department of Energy, Office of Science User Facility. The work at the University of Maryland (X.Z., I.T.) was supported in part by the National Institute of Standards and Technology Cooperative Agreement 70NANB17H301 and the Center for Spintronic Materials in Advanced infoRmation Technologies (SMART) one of the centers in nCORE, a Semiconductor Research Corporation (SRC) program sponsored by NSF and NIST. A.N.M. acknowledges the Target Program of Basic Research of the National Academy of Sciences of Ukraine "Prospective basic research and innovative development of nanomaterials and nanotechnologies for 2020 - 2024", Project № 1/20-H (state registration number: 0120U102306) and the European Union's Horizon 2020 research and innovation program under the Marie Skłodowska-Curie (grant agreement No 778070). This manuscript has been authored by UT-Battelle, LLC, under Contract No. DE-AC0500OR22725 with the U.S. Department of Energy. The United States Government retains and the publisher, by accepting the article for publication, acknowledges that the United States Government retains a non-exclusive, paid-up, irrevocable, world-wide license to publish or reproduce the published form of this manuscript, or allow others to do so, for the United States Government purposes. The Department of Energy will provide public access to these results of federally sponsored research in accordance with the DOE Public Access Plan (http://energy.gov/downloads/doe-public-access-plan).

## Author contributions
S.V.K. designed and led the project. S.V.K., C.T.N., R.K.V., M.Z., contributed to the Bayesian analysis code and data analysis. C.T.N. performed STEM experiments and data parameterization. X.Z. and I.T. synthesized the BFO thin films. E.A.E. and A.N.M. contributed the ferroelectric domain wall models. S.V.K., C.T.N., and R.K.V. prepared the manuscript.

## Competing interests
The authors declare no competing interests.
