## [Peer Review File · Nature Communications]

REVIEWER COMMENTS

Reviewer #1 (Remarks to the Author):

The authors present a study on the Bayesian analysis of HAADF-STEM data for domain walls in BiFeO₃ ferroelectrics. The data here is of high quality. This is a new exploration on the extractions of potential physics behind TEM data, which may have implications for studies of other materials systems.

Since this is an application of statistical method for analysis of TEM data, my major concern is that how to ensure the sampling process is not biased? The authors have used BiFeO₃ films grown on SrTiO₃ substrate, and according to Figure 2a, the interface seems to be coherent, thus the considerable mismatch in this system should exert large strain on BiFeO₃ films. This strain may affect the domain wall structures. For example, the 109 wall and 180 wall here in Figure 2a are actually in an irregular shape, although the authors try hard to choose the ROI.

The descriptions of STEM imaging and related analysis of domain walls in ferroelectrics, in the introduction section, are basically good, although some very related study seems missing here, for examples, the Bloch-type polarization components in PbTiO₃ ferroelectrics by first-principles calculations (Journal of Applied Physics 116, 224105 (2014)).

Some discussions and suggestions:

- 1: Figure 2a is incorrectly referred as Figure 1a, as seen in page 8, 9 and 11.
- 2: The "ROI" abbreviation should be designated when it appears first time to readers.
- 3: "However, for the analysis of the experimental data we treat the phenomenological wall parameters as independent." What is the relationships between these phenomenological parameters? Is this treatment reliable? The authors should discuss more about it.
- 4: "Hence, the fitting function is expanded to include a vertical polarization offset term, P_0 ." However, bias is obvious in the original image as observed by the authors. I suggest the authors should discuss more and rationalize such a treatment.

X.L.Ma

Reviewer #2 (Remarks to the Author):

The authors apply Bayesian analysis on STEM data to infer nature of domain walls in a BiFeO₃ thin films. The authors state clear goals, e.g., to differentiate between three physical models describing domains in materials with first and second order phase transitions, and have even more ambitious goal to ultimately establish requirements on the measurement technique which would allow such differentiation possible. These are important goals, original, have potential to advance the field and the paper is in this sense very valuable.

As the authors mention in introduction "... its (Bayesian theory) adoption by the basic science fields has been slow." Thus authors are aware that not many researchers are familiar with the details of the Bayesian analysis. When I accepted to review this rare paper which applies Bayesian theory in the field of ferroelectrics, I was hoping that it would be informative and help me understand the approach. Unfortunately, this is not the case. This is obviously not the the fault of the authors, but it does hint that there will be other readers like me who will have more questions than answers after reading the paper.

I will refrain from commenting paper that is for me nontransparent, assumes rather advanced practical knowledge of the Bayesian analysis and which I mostly did not understand. I hope the other reviewers be of more help.

I will only make few minor comments:

-I could not make "link to notebook" active and could not see this notebook.

-authors wrote (twice) "We show that the domain wall shape is sensitive to the material physics, including the nature of the order parameter(s), form of the GLD free energy, and numerical value of corresponding parameters." I do not see how domain wall shape can be sensitive to "form of the GLD theory". This sounds odd. This is possible if the authors are considering only theoretical models of domains, but since they also have experimental results the statement sounds strange as it implies that an experimental object (domain wall) would depend on parameters of a theory that describes it. Or have I misunderstood something?

-Authors wrote "Elastic fields, which are, in fact, the secondary order parameters, satisfy equation of state and mechanical equilibrium equations, whereas the strains and/or stresses should be defined at the system boundaries." I do not understand what is the difference between "elastic fields" and "strains and stresses"?

-If I understood correctly the authors could not, based just on experimental data differentiate between different models (They wrote at the end of the article "For the specific case of 109° and 180° domain walls in BiFeO_3 , the combination of sampling and noise preclude a reliable differentiation of physical mechanisms.") Why did not authors use a system on which this could be accomplished, considering that they are aware that "(Bayesian theory)c adoption by the basic science fields has been slow"? Why choose a material where this analysis does not work very well?

-Finally, the authors wrote "Following the seminal work by the Ramesh group demonstrating enhanced conductivity at the ferroelectric wall...". This paper has 19 coauthors and they do not all belong to one group. This sentence should be rephrased. More importantly, other groups have discovered earlier that domain walls in a ferroic material can have different conductivity than domains. See for example A. Aird and E. Salje, "Sheet superconductivity in twin walls," J. Phys.: Condens. Matter, vol. 10, p. L377, 1998.

Response to reviewer 1:

The authors present a study on the Bayesian analysis of HAADF-STEM data for domain walls in BiFeO₃ ferroelectrics. The data here is of high quality. This is a new exploration on the extractions of potential physics behind TEM data, which may have implications for studies of other materials systems.

We thank the reviewer for examining the manuscript, and his/her high opinion of this work.

Since this is an application of statistical method for analysis of TEM data, my major concern is that how to ensure the sampling process is not biased? The authors have used BiFeO₃ films grown on SrTiO₃ substrate, and according to Figure 2a, the interface seems to be coherent, thus the considerable mismatch in this system should exert large strain on BiFeO₃ films. This strain may affect the domain wall structures. For example, the 109 wall and 180 wall here in Figure 2a are actually in an irregular shape, although the authors try hard to choose the ROI.

The reviewer raises a set of very interesting questions. In short, because our aim is employing Bayesian analysis to derive system physics via the parameters and fitness of several domain wall models, we constrain the sampling to regions where these models are expected to be most valid. Common 1-D or periodic boundary domain wall models including those used here correspond closest to long-range uniform domain walls in experimental systems. The selected ROIs best satisfy the model criteria within the datasets, being domain wall segments on known rhombohedral BiFeO₃ equilibrium 109° and 180° domain wall planes without visible step edges. Including ill-suited regions will knowingly introduce confounding factors, such as off-axis rotations or domain wall roughness, resulting in non-quantified artifacts (e.g. broadening) in the projected structure not captured by the model symmetry. This treatment also has a significant practical advantage in that the geometric complexity is reduced to a simple series of nominally independent atom-wide domain wall “samples” subject to statistical analysis.

The reviewer is correct in noting the domain structure irregularity, the electron-transparently thinned thin film cross sections being highly sensitive to boundary conditions as well as internal defect chemistry. Only relatively short segments of the domain walls exhibited unknicked, on-edge, equilibrium-plane structures and thus were the focus of our study. As the reviewer points out epitaxial strain effects are certainly present, as manifest by the density of observed ferroelastic-type domains walls (e.g. 109°), and will contribute to the observed domain structure. While these extrinsic factors are not expected to induce significant microstructural deviations from the bulk-like rhombohedral phase within the measurement error of the instrument, our simple models are not predicated on this and should be valid in this regime. It actually highlights one of the key goals of this type of analysis: given a practical system such as the given SrTiO₃/BiFeO₃ thin film geometry can local physics (e.g. 1st vs 2nd order phase transition) and corresponding confidence level be determined (from a Bayesian analysis of local structure with comparative models)?

To the broader spirit of the representativeness of the sampling and the robustness of the Bayesian analysis of the domain walls we have repeated the treatment for a new dataset obtained from the same sample. The dataset is shown in Fig 1 and consists of another pair of 109° and 180° ROIs.

One difference of this new dataset is the close proximity of a misfit-strain relaxing dislocation pair (yellow arrows) so the local strain state is not identical but does provide a counter example where the epitaxial strain is absent. For easy reference, the manuscript figures are paired below with the corresponding data derived from the new dataset. We have also repeated the analysis for a further subsampling of the dataset(s), restricted to the region near (within 5 atoms) the domain wall, and included the WAIC weights for comparison in Table 1.

Fig 1. STEM-HAADF (left) and colored polar displacement images (right) of the original (#1, top) and additional (#2, bottom) dataset. Pair of misfit edge-dislocations are indicated by yellow arrows, creating plotting artifacts due to grid-addressing failure. The top of the image in dataset #2 is tilted off axis, resulting in errors/noise in atom position fitting (grainy pink regions at top)

Both datasets are qualitatively alike with domain wall widths narrow down to the level of the discretization of the atomic lattice (Figs 2 & 3) and a correspond high level of uncertainty between models for all cases as indicated by the comparative WAIC weights (Table 1).

Parameter posterior distributions are generally in very good agreement, especially for the 109° domain wall case. Please note, several parameters (x_0 , P_S , and P_0) will inherently vary between regions and their axes offsets can be disregarded.

Fig 2. Experimental 109° Domain walls (top), corresponding models (middle), and posterior distributions (bottom) for datasets #1 (left) & #2 (right).

Fig 3. Experimental 180° Domain walls (top), corresponding models (middle), and posterior distributions (bottom) for datasets #1 (left) & #2 (right).

For the 109° domain walls (Fig 2) there is high quantitative agreement between datasets as illustrated by the comparative posterior & joint distributions (Fig. 2), as well as the WAIC scores (Table 1). Note, dataset 2 presented with a fixed offset so a P_0 offset term was added as was done for the original 180° domain wall. The 180° domain walls exhibit some minor differences between the datasets with a lower L_c and higher η predictions for the second case (Fig. 3) but similar distributions otherwise.

(Original)
Data: full ROI width

Data: within 5 atomic columns of
the domain wall

Dataset #1

Model	109° wall		180° wall		
	Rank	Weight	Rank	Weight old	Weight
Model I	1	0.49	1	0.82	0.56
Model II	3	0.22	3	0.18	0.11
Model III	2	0.29	2	0.00	0.33

Model	109° wall		180° wall	
	Rank	Weight	Rank	Weight
Model I	1	0.39	1	0.45
Model II	3	0.3	2	0.23
Model III	2	0.3	3	0.32

Dataset #2

Model	109° wall		180° wall	
	Rank	Weight	Rank	Weight
Model I	1	0.54	2	0.37
Model II	2	0.28	1	0.38
Model III	3	0.18	3	0.25

Model	109° wall		180° wall	
	Rank	Weight	Rank	Weight
Model I	2	0.34	2	0.31
Model II	3	0.29	1	0.46
Model III	1	0.37	3	0.23

Table 1. WAIC weights of the 109° and 180° domain walls for the three models. Original dataset is top, dataset #2 is bottom. Left is for the original analysis using the entire ROI width shown in Fig 1, right is the near-domain wall region only. Note, for the original case our first reported values are in red but we have a correction of 180° domain wall WAIC values beside it in black (an uncaught error)

As in our original analysis, a comparison of the models of WAIC weights for all settings affirms a high degree of uncertainty between models. Notably, further restricting the data to the near-wall atoms (Table 1, right side) tends to push the models to a closer tie. A correction of an uncaught error in our initial WAIC scores for the 180° domain wall case has the same effect. While the initial dataset still gives an edge to model 1, the 2nd dataset shows no consistent preference.

The descriptions of STEM imaging and related analysis of domain walls in ferroelectrics, in the introduction section, are basically good, although some very related study seems missing here, for examples, the Bloch-type polarization components in PbTiO₃ ferroelectrics by first-principles calculations (Journal of Applied Physics 116, 224105 (2014)).

We have added this suggested reference.

Some discussions and suggestions:

1: Figure 2a is incorrectly referred as Figure 1a, as seen in page 8, 9 and 11.

Well spotted, thank you!

2: The “ROI” abbreviation should be designated when it appears first time to readers.

We have added this designation.

3: “However, for the analysis of the experimental data we treat the phenomenological wall parameters as independent.” What is the relationships between these phenomenological parameters? Is this treatment reliable? The authors should discuss more about it.

In this case, the statement simply implies that the parameters as used in the Bayesian fit are independent.

4: "Hence, the fitting function is expanded to include a vertical polarization offset term, P_0 ." However, bias is obvious in the original image as observed by the authors. I suggest the authors should discuss more and rationalize such a treatment.

We have rephrased this and the preceding line for (hopefully) more clarity to:

“There is an observable asymmetry with P not centered around zero, a known artifact of small off-axis misalignments which manifests in a similar polarization asymmetry.⁵² Since we are concerned only with the polarization delta of the domain wall, this component is treated with a fixed offset term, P_0 , added to the fitting function.”

Ambiguity in projected centro-symmetry breaking due to axial misalignment between the local zone axis and electron beam direction is a well-known effect in the (S)TEM community, covered well for this context by the cited reference (Liu, Y. et al., *Journal of Materials Research* **32**, 947-956 (2017)). Fortunately, as we are concerned with a polarization delta occurring at domain walls it's easily isolated from the background, actually not altogether different from polarization measurements experimentally derived from ferroelectric switching currents instead of any direct measure of the absolute polarization.

Reviewer #2:

The authors apply Bayesian analysis on STEM data to infer nature of domain walls in a BiFeO₃ thin films. The authors state clear goals, e.g., to differentiate between three physical models describing domains in materials with first and second order phase transitions, and have even more ambitious goal to ultimately establish requirements on the measurement technique which would allow such differentiation possible. These are important goals, original, have potential to advance the field and the paper is in this sense very valuable.

We thank the reviewer for very high opinion of the manuscript.

As the authors mention in introduction "... its (Bayesian theory) adoption by the basic science fields has been slow." Thus authors are aware that not many researchers are familiar with the details of the Bayesian analysis. When I accepted to review this rare paper which applies Bayesian theory in the field of ferroelectrics, I was hoping that it would be informative and help me understand the approach. Unfortunately, this is not the case. This is obviously not the fault of the authors, but it does hint that there will be other readers like me who will have more questions than answers after reading the paper.

We very much appreciate the reviewer's candid perspective here. We have expanded the Bayesian inference section of the introduction (final paragraph) with the aim to benefit unfamiliar readers, elaborated some of the descriptions (e.g. the WAIC analysis), and have added some references to helpful texts.

I will refrain from commenting paper that is for me nontransparent, assumes rather advanced practical knowledge of the Bayesian analysis and which I mostly did not understand. I hope the other reviewers be of more help.

We appreciate the overall positive opinion of the reviewer, and we do agree that Bayesian analysis is a very powerful and promising, but yet unconventional, method.

I will only make few minor comments:

-I could not make "link to notebook" active and could not see this notebook.

Apologies to the referee, this was not an active link pending a host location. Please find an interim link here if you would like to view the python notebook:

https://colab.research.google.com/drive/1Hq5RyEH0g9beCq844z9jDSF_qlZoYJi3?usp=sharing

-authors wrote (twice) "We show that the domain wall shape is sensitive to the material physics, including the nature of the order parameter(s), form of the GLD free energy, and numerical value of corresponding parameters." I do not see how domain wall shape can be sensitive to "form of the GLD theory". This sounds odd. This is possible if the authors are considering only theoretical models of domains, but since they also have experimental results the statement

sounds strange as it implies that an experimental object (domain wall) would depend on parameters of a theory that describes it. Or have I misunderstood something?

We apologize for the unfortunate wording. We mean “profile” by shape. Here, we utilize this theoretical dependence of domain wall profile on materials physics to learn to which extent the theoretical model parameters can be learnt from experimental data. For clarity, the text has been modified to read: “Predicted domain wall shapes are dependent on parameters of the underpinning Landau theory. Under the assumptions of the validity of the theory, this then suggests that the domain wall profiles observed allow inversion to yield the parameters of the underlying Landau model, and thus, infer the order of the phase transition.”

-Authors wrote “Elastic fields, which are, in fact, the secondary order parameters, satisfy equation of state and mechanical equilibrium equations, whereas the strains and/or stresses should be defined at the system boundaries.” I do not understand what is the difference between “elastic fields” and “strains and stresses”?

We again apologize for unfortunate wording. Here, we mean that strain and stress are local variables; however, the corresponding fields are determined by the boundary conditions, i.e. are non-local.

-If I understood correctly the authors could not, based just on experimental data differentiate between different models (They wrote at the end of the article "For the specific case of 109° and 180° domain walls in BiFeO3, the combination of sampling and noise preclude a reliable differentiation of physical mechanisms.") Why did not authors use a system on which this could be accomplished, considering that they are aware that "(Bayesian theory) adoption by the basic science fields has been slow"? Why choose a material where this analysis does not work very well?

The reviewer is not mistaken, this comparative analysis did not differentiate the three models with confidence, seeing only a slight preference for model 1. Certainly, a more definitive outcome would better showcase application of Bayesian theory, however this is a case of a scientific question (local nature of ferroelectric physics) in search of a tool rather than vice versa. It is a promising material system for such an analysis using atomic STEM data, with well resolved atomically defined features corresponding to a ferroic order parameter and clean physical models. Even in this case, the Bayesian analysis provides a quantification of the model uncertainty, parameter probability distributions, and predictions for the experimental limits wherein models can be distinguished. Moreover, it is our view this application to STEM data is extremely promising, limited here only by the discreteness of the atomic-lattice relative to the model feature in question. We foresee it as a viable universal approach for less discrete order parameter gradients (i.e. most of them), including examples within this materials system such as for strain and oxygen octahedral tilt gradients at junctions.

-Finally, the authors wrote "Following the seminal work by the Ramesh group demonstrating enhanced conductivity at the ferroelectric wall...". This paper has 19 coauthors and they do not all belong to one group. This sentence should be rephrased. More importantly, other groups

have discovered earlier that domain walls in a ferroic material can have different conductivity than domains. See for example A. Aird and E. Salje, "Sheet superconductivity in twin walls," J. Phys.: Condens. Matter, vol. 10, p. L377, 1998.

We thank the reviewer for this input, and have adjusted the manuscript correspondingly.

REVIEWER COMMENTS

Reviewer #1 (Remarks to the Author):

I find the authors have addressed my major concerns by providing new data and analysis. But after reading the whole text of the revised manuscript, I still have to raise some concerns, particularly on the contextualization in the introduction section.

In reviewing STEM characterization of the internal structure of ferroelectric domains in the past decade by the introduction and proliferation of atomic-resolution spherical aberration corrected microscopes which allow direct observations of the ferroelectric domain wall (and other interfaces) on the atomic level, two important studies are mentioned, such as Jia et al. in *Science* (2011) and Yadav et al. in *Nature* (2016). However, I have to comment that when talking about the *Science* (2011) paper, another paper by C. T. Nelson et al. (*Nano Letters*, 2011) should not have been missed (this work was done by the same author of the present NC manuscript). These two papers are equally first in atomic mapping of polarizations in real space by aberration-corrected electron microscope.

Similarly, when talking about the *Nature* (2016) work, one should not miss the earlier study (Y.L. Tang et al. *Science* 2015) on which the *Nature* (2016) paper is based. One can see that the sample system in the *Nature* (2016) paper is the same as that in the *Science* paper (2015), and the domain configurations are similar to those in the *Science* paper (2015). By using similar strategy, recently a new kind ferroelectric domain and its array is also reported (Y. J. Wang et al., *Nature Materials* 2020).

In high-profile journals like *Nature Communications*, a fair and reasonable contextualization on the previous studies in the field is needed, which can also be regarded as a contribution.

In addition, in the response to the reviewers' concerns, the authors mentioned that they involved the JAP paper (116, 224105 (2014)) in the text, however, I do not see it in reference list.

Reviewer #2 (Remarks to the Author):

The authors have made recommended changes and have addressed my questions. I can now recommend the paper for publication.

Response to reviewer 1:

I find the authors have addressed my major concerns by providing new data and analysis. But after reading the whole text of the revised manuscript, I still have to raise some concerns, particularly on the contextualization in the introduction section.

In reviewing STEM characterization of the internal structure of ferroelectric domains in the past decade by the introduction and proliferation of atomic-resolution spherical aberration corrected microscopes which allow direct observations of the ferroelectric domain wall (and other interfaces) on the atomic level, two important studies are mentioned, such as Jia et al. in Science (2011) and Yadav et al. in Nature (2016). However, I have to comment that when talking about the Science (2011) paper, another paper by C. T. Nelson et al. (Nano Letters, 2011) should not have been missed (this work was done by the same author of the present NC manuscript). These two paper are equally first in atomic mapping of polarizations in real space by aberration-corrected electron microscope.

Similarly, when taking about the Nature (2016) work, one should not miss the earlier study (Y.L. Tang et al. Science 2015) on which the Nature (2016) paper is based. One can see that the sample system in the Nature (2016) paper is the same as that in the Science paper (2015), and the domain configurations are similar to those in the Science paper (2015). By using similar strategy, recently a new kind ferroelectric domain and its array is also reported (Y. J. Wang et al., Nature Materials 2020).

In high-profile journals like Nature Communications, a fair and reasonable contextualization on the previous studies in the field is needed, which can also be regarded as a contribution.

In addition, in the response to the reviewers' concerns, the authors mentioned that they involved the JAP paper (116, 224105 (2014)) in the text, however, I do not see it in reference list.

We are happy to expand the citation list of HRSTEM works of ferroelectric systems. We have added the following references which includes those suggested by the reviewer.

- 36 Nelson, C. T. et al. Spontaneous Vortex Nanodomain Arrays at Ferroelectric Heterointerfaces. *Nano Letters* **11**, 828-834 (2011).
- 37 Tang, Y. L. et al. Observation of a Periodic Array of Flux-Closure Quadrants in Strained Ferroelectric Pbtio3 Films. *Science* **348**, 547 (2015).
- 38 Mundy, J. A. et al. Atomically Engineered Ferroic Layers Yield a Room-Temperature Magnetoelectric Multiferroic. *Nature* **537**, 523-527 (2016).
- 39 Zeches, R. J. et al. A Strain-Driven Morphotropic Phase Boundary in Bifeo3. *Science* **326**, 977 (2009).
- 40 Borisevich, A. Y. et al. Suppression of Octahedral Tilts and Associated Changes in Electronic Properties at Epitaxial Oxide Heterostructure Interfaces. *Physical Review Letters* **105**, 087204 (2010).
- 41 Das, S. et al. Observation of Room-Temperature Polar Skyrmions. *Nature* **568**, 368-372 (2019).
- 42 Wang, Y. J. et al. Polar Meron Lattice in Strained Oxide Ferroelectrics. *Nature Materials* **19**, 881-886 (2020).

We thank the reviewer for catching the missing citation! Not sure if it was by mistake or error but it didn't propagate to the final revised draft. It has been corrected.

Response to reviewer 2:

The authors have made recommended changes and have addressed my questions. I can now recommend the paper for publication.

We are grateful to the reviewer for their recommendation of this work.

REVIEWERS' COMMENTS

Reviewer #1 (Remarks to the Author):

The authors have well addressed my concerns, and I recommend acceptance of publication of this revised paper.

X.L.Ma